# Smartphone-Based Rapid Quantitative Detection Platform with Imprinted Polymer for Pb (II) Detection in Real Samples

**DOI:** 10.3390/polym16111523

**Published:** 2024-05-28

**Authors:** Flor de Liss Meza López, Christian Jacinto Hernández, Jaime Vega-Chacón, Juan C. Tuesta, Gino Picasso, Sabir Khan, María D. P. T. Sotomayor, Rosario López

**Affiliations:** 1Technology of Materials for Environmental Remediation (TecMARA) Research Group, Faculty of Sciences, National University of Engineering, Lima 15333, Peru; flor.meza.l@uni.pe (F.d.L.M.L.); jvegac@uni.edu.pe (J.V.-C.); gpicasso@uni.edu.pe (G.P.); 2Laboratory of Instrumental Analysis Environment, Faculty of Sciences, National University of Engineering, Av. Tupac Amaru 210, Lima 15333, Peru; christian@uni.edu.pe; 3Laboratorio de Biotecnología, Universidad Nacional Autónoma de Alto Amazonas, Calle Prolongación, Libertad 1220, Yurimaguas 16501, Peru; jtuesta@unaaa.edu.pe; 4Department of Exact Sciences and Technology, State University of Santa Cruz, Ilhéus 45662-900, BA, Brazil; skhan@uesc.br; 5Institute of Chemistry, State University of São Paulo (UNESP), Araraquara 14801-970, SP, Brazil; 6National Institute for Alternative Technologies of Detection, Toxicological Evaluation and Removal of Micropollutants Radioactives (INCT-DATREM), Araraquara 14801-970, SP, Brazil

**Keywords:** smartphone, ion-imprinted polymer (IIP), Pb^2+^, colorimetric sensor

## Abstract

This paper reports the successful development and application of an efficient method for quantifying Pb^2+^ in aqueous samples using a smartphone-based colorimetric device with an imprinted polymer (IIP). The IIP was synthesized by modifying the previous study; using rhodizonate, 2-acrylamido-2-methylpropane sulfonic acid (AMPS), *N*,*N*′-methylenebisacrylamide (MBA), and potassium persulfate (KPS). The polymers were then characterized. An absorption study was performed to determine the optimal conditions for the smartphone-based colorimetric device processing. The device consists of a black box (10 × 10 × 10 cm), which was designed to ensure repeatability of the image acquisition. The methodology involved the use of a smartphone camera to capture images of IIP previously exposed at Pb^2+^ solutions with various concentrations, and color channel values were calculated (RGB, YMK HSVI). PLS multivariate regression was performed, and the optimum working range (0–10 mg L^−1^) was determined using seven principal components with a detection limit (LOD) of 0.215 mg L^−1^ and R^2^ = 0.998. The applicability of a colorimetric sensor in real samples showed a coefficient of variation (% RSD) of less than 9%, and inductively coupled plasma mass spectrometry (ICP–MS) was applied as the reference method. These results confirmed that the quantitation smartphone-based colorimetric sensor is a suitable analytical tool for reliable on-site Pb^2+^ monitoring.

## 1. Introduction

Pollution generated by heavy metals, as a result of activities such as industrialization and mining, has been considered a global environmental problem that is increasing all over the planet [1]. Lead is an important compound used as an intermediate in processing industries such as plating, paint and dye, and lead batteries [2]. Divalent lead (Pb^2+^) has severe toxicity and is generally assumed to be one of the most dangerous metals due to its highly harmful multisystem effect, especially in children [1,2,3]. Through the soil–plant–animal–human food chain system, Pb^2+^ is transferred to animals and humans [2]. Therefore, the development of reliable methods for the quantification of lead in various matrices is of particular importance.

Different methods have been developed to quantify this metal’s presence in different sources [3,4,5]. Some investigations that have attracted a significant amount of attention so far are those that have developed some analysis platforms to quantitatively detect Pb^2+^, such as colorimetric [6], spectroscopic [3], fluorescent [7,8,9], and electrochemical [5,10] sensors. In such devices, the optical variation that is controlled can be the color variation, through reflectance or transmittance; the refractive index; evanescent wave; or SPR (surface plasmon resonance) variation [5,6,7,8,9,10]. However, most of these methods need to be performed in the laboratory with professional instruments using complex operations and cannot be universally applied to daily life. Therefore, the development of a portable, easy-to-use, fast, and efficient quantification method for Pb^2+^ is of great importance.

The quantification efficiency in technologies based on optical sensor platforms is based on the recognition reagent used [9,10,11], so its correct choice is crucial. Optical type sensors provide an optical response depending on the concentration of the analyte in a sample and depending on the optical property that has been measured: absorbance, reflectance, fluorescence, phosphorescence, luminescence, Raman scattering, evanescence, refractive index, etc. [11].

The colorimetric interaction between Pb^2+^ and some reagents has generated materials whose application also involves color changes, which allows for colorimetric quantification [6,12,13,14,15,16]. One of the reagents that has shown a strong interaction with Pb^2+^ is sodium rhodizonate. The use of sodium rhodizonate in criminal analysis is very common and is considered a very sensitive and specific test [12]. The interaction between sodium rhodizonate and lead is based on the formation of a colored complex (reddish coloration) by resonance, where two rhodizonate molecules form a ring around the lead and the oxygens of the rhodizonate enol groups complex the metal [13]. In the literature, there are a few precedents for the use of rhodizonate in the detection of lead [13,16], as well as the use of rhodizonate for the detection of Pb^2+^ in water samples using adsorbent papers impregnated with the reagent [6]. More recent studies indicate that the interaction between rhodizonate and lead is very strong and specific due to the predetermined orientation of the coordination geometry generated by the rhodizonate ligand [6,13,16]; thus, a prevalence of ion chelation is observed in Pb^2+^ in the presence of other metals in the colorimetric detection of Pb (II).

However, direct analysis is often disturbed by the presence of a complex matrix in environmental samples [6,11]. Consequently, the selective separation of Pb^2+^ from natural samples needs much more attention. Among the various options to overcome selectivity difficulties in the analyte quantification process, the use of so-called imprinted ionic polymers (IIP) is proposed. The main advantage of this type of material is selective recognition through the formation of cavities that are complementary in size and shape to the analyte (metal ion) [16,17,18,19].

The possibility of direct application of this type of colorimetric sensor for in situ monitoring of lead in different matrices, conditioned on sensor platforms that can be implemented for this purpose, is raised. Previously developed and studied IIPs are particularly suitable for designing portable colorimetric detection platforms due to their negligible background color and reflectance emission, large charge capacity, and proven recognition efficiency [20,21,22,23]. The polymeric material can provide an inert environment to ensure that the encapsulated and immobilized targeting material has good stability [11,20,22].

On the other hand, the advent of universal digital imaging technology offers new possibilities for simple colorimetric detection. With the help of digital imaging technology, a colorimetric sensor becomes more real, simple, efficient, and sensitive in quantification [24]. At present, the use of portable devices such as SMARTPHONE for real-time monitoring has become an area of great interest [24,25,26,27,28,29,30,31,32,33,34,35]. A proper synergy between the smartphone and the sensor is very important because the sensor will collect the data and then the mobile phone software can process the data and generate readable information [28,29,30]. The colorimetric quantification mechanism focuses mainly on color patterns generated by the interaction between the recognition material and the analyte. RGB (red, green, blue value) analysis is one of the most widely used analysis techniques for imaging data. Red, green, and blue values can be obtained by analyzing digital photographs with specific software, and therefore, RGB analysis has been widely used in the field of rapid, portable, and quantitative detection in recent years [26,29,30,31]. In addition, other color spaces are also used, such as the following: cyan, magenta, yellow, and key (black) (CMYK) as well as hue, saturation, and value (HSV) [32]. Almost all optical analyses can be integrated with smart platforms, including absorbance, fluorescence [24,27,31], reflectivity [25], surface plasmon resonance (SPR) [27], and bioluminescence [30]. Detection methods based on RGB analysis using SMARTPHONE–like smart platforms do not require expensive instruments, making on–site detection and remote transmission and analysis of the results possible.

We take into account the favorable properties of IIP’s polymeric materials and the high specificity of sodium rhodizonate as a colorimetric sensor for the detection of Pb^2+^, it is possibly to measure Pb^2+^ ions. By implementing SMARTPHONE intelligent platforms and RGB analysis, quantitative detection can be conveniently achieved, as can the application of analytical devices for efficient and real-time monitoring of environmental samples of interest. Rhodizonate forms a colored complex in the presence of lead by resonance, resulting in the color change of the IIP polymeric material. The quantification method would include capturing images of the reference sample and the target sample with the smartphone camera. Then, the images would be digitized and decomposed into different color spaces that will be decided according to specific test conditions. The result could be obtained by comparing the target color, through software, with the reference by applying various methods to ensure a reliable result. Based on this principle, the research would be a pioneer in the application of imprinted ionic polymers incorporated in SMARTPHONE devices for the quantification of heavy metals. We intend to provide a system for optical quantification of Pb^2+^ that meets all the requirements of an analytical sensor, including good sensitivity and selectivity in the presence of competitive ions. Moreover, it is a convenient, non–destructive, and portable system that is easy to handle to monitor various matrices of interest.

## 2. Materials and Methods

All chemicals used in this study were of analytical or HPLC grade and were purchased from Sigma Aldrich^®^ (St-Quentin Fallavier, France): rhodizonic acid disodium salt (rhodizonate), lead nitrate (Pb(NO_3_)_2_), 2-acrylamido-2-methylpropane sulfonic acid (AMPS), *N*,*N*-methylene bisacrylamide (MBA), potassium persulfate (KPS), HNO_3_ (99.999%), HCl (37%), and NaOH. Stock solution of Pb^2+^ (1000 mg L^−1^) and multi-element standard solution (100 mg L^−1^) were also used. All solutions were prepared in deionized water (18 MΩ cm at 25 °C). 

### 2.1. Synthesis of Ion Imprinted Polymer (IIP)

The polymeric material was synthesized in a covered flask with a capacity of 100 mL. First, 0.01 mmol of rhodizonate and 0.06 mmol of AMPSA were dissolved in 50 mL of ultrapure water under vigorous stirring for 120 min and in an inert environment by purging with N_2_. Afterward, Pb(NO_3_)_2_ was added to the mixture; the reaction mixture was left for 6 h at room temperature. Next, the cross-linker and radical initiator (0.6 mmol of MBA and 0.01 mmol of KPS, respectively) were added to carry out the polymerization at 70 °C for 24 h in an N_2_ environment. The obtained polymeric material was rinsed, dried, and leached with a solution of 1.0 mol L^−1^ HNO_3_ under constant agitation and repeated washing until the complete elimination of the metal ion was achieved. The analysis of the Pb^2+^ ion was performed by flame atomic absorption spectrophotometry (FAAS) at *λ* = 217.0 nm, using a Savant AA GBC spectrometer. For purposes of comparison, the non-polymeric material (NIP) was synthesized following the same procedure; the materials were denoted by IIP-AMPS and NIP-AMPS.

The resulting materials were characterized by infrared spectroscopy in ATR mode using a Bruker Vertex 70 spectrophotometer, and the morphology of materials by scanning electron microscopy (SEM) using a JEOL JSM 6330F microscope. The thermal stability was assessed via thermogravimetric analysis using a PerkinElmer STA 6000 instrument, the textural properties by BET method based on N_2_-sorption isotherms using a Micrometrics GEMINI VII 2390t instrument.

### 2.2. Adsorption Test

Adsorption tests were performed in a batch system, and the pH of the medium and the mass of the adsorbent influence were analyzed. Adsorption kinetic experiments were performed with optimum conditions to determinate the optimal time of adsorption. In addition, the adsorption isotherm experiment was carried out by adding 10 mg of polymer IIP or NIP to 10 mL of lead solutions with different concentrations in order to determinate the equilibrium adsorption capacity (q_e_, mg g^−1^) using Equation (1).
(1)$$q_{e}=\frac{(C_{o}-C_{e})V}{m}$$
where C_0_ and C_e_ are the initial and equilibrium concentration of Pb^2+^ (mg L^−1^), respectively; m represents the mass of the polymer (g); and V is the volume of the Pb^2+^ solution (L). The adsorption selectivity was evaluated by testing in mixed solutions, such as Pb^2+^, Cu^2+^, Ni^2+^, Cd^2+^, Zn^2+^, Ca^2+^, Fe^2+^, and Hg^2+^ ion solutions, and real samples were evaluated. 

The reuse of adsorbent IIP–AMPS was investigated to evaluate its efficiency and practicability. The imprinted polymer obtained had the capacity for regeneration through successive leaching with HNO_3_ until achieving the complete release of the metal ion from the polymer structure. For this purpose, the solids were recovered and washed with 1M HNO_3_ solutions until the solution remaining from the washing indicated the absence of the metal ion. The evaluation of the metal in solution was carried out by atomic absorption spectroscopy (FAAS). The regeneration of the material in each cycle involved 6 washing processes. After this process, the material was reused for subsequent adsorption process, continuing with another regeneration cycle.

### 2.3. Image Acquicition and SMARTPHONE Procesing

To obtain the images, for subsequent processing and construction of the Pb^2+^ calibration curve with a smartphone application, the accessory was obtained using a Flashforge 3D printer Flashforge with Creator 3 software. The device consisted of a black box of dimensions of 10 × 10 × 10 cm, 1 cm thickness which was designed to ensure repeatability of image adquicition (Figure 1a), and the sample holder (Figure 1b) was designed to hold several samples at a time. 

For image capture, a Samsung Galaxy A54 5G smartphone, model SM–A546E, with a 50, 12.5, and 5 MP rear high-resolution triple camera was used (important for obtaining images with good quality and resolution). The camera sensor’s sensitivity was set to automatic ISO with optical image stabilization. The sample holder containing the samples from the adsorption isotherm study (the solids were recovered after each test carried out and were dried on silica gel in an oven at 60 °C) was photographed and the images were processed with a program to obtain RGB, YMK, and HSV values for the construction of the calibration curves through a multivariate PLS analysis (partial least square regression).

### 2.4. Study of Figures of Merit

The colorimetric sensors were evaluated under the previously established optimal conditions, aiming at estimating the limit of detection (LOD) and the limit of quantification (LOQ) of the proposed method. Analytical curves were constructed using polymeric solids containing Pb^2+^ concentrations ranging from 1 to 10 mg L^−1^. The LOD and LOQ were calculated as follows:(2)$$LOD=\frac{3S}{\alpha }$$
(3)$$LOQ=\frac{10S}{\alpha }$$
where S is the standard deviation of the analytical signal corresponding to the blank, and *α* is the slope of the analytical curve. Repeatability and reproducibility tests of the proposed sensor were also conducted.

### 2.5. Applicability

The applicability of the colorimetric sensor device in environmental water samples was evaluated. Samples from Peruvian rivers were filtered through 0.45 µm membrane filters adjusted to a pH approximately 6.0 ± 0.1 and fortified with a known amount of Pb^2+^ solution. Batch adsorption was evaluated at the optimal experimental conditions, and the polymeric solids were recovered and analyzed using an imaging accessory to assess their coloration and calculate the concentrations based on a previously obtained calibration curve.

## 3. Results and Discussions

### 3.1. Synthesis of Polymeric Material (IIP)

The IIP prepared followed a modified methodology of the previous proposal [19]. According to previous studies [13], instability of the rhodizonate ligand against light and under basic solution conditions is evident. To prevent this condition from affecting the polymerization performance and, therefore, the adsorption capacity of the polymeric material, modifications were considered, such as keeping the container completely covered to avoid light incidence in any way. We maintained the inert environment throughout the synthesis process, bubbling with nitrogen gas at each stage.

Changes in the solution containing the dissolved rhodizonate ligand exposed to the indicated instability conditions were verified by mass spectrometry (MS). The analyses were carried out in direct infusion mode using the source electrospray ionization system (DI–ESI) and the following parameters: mass spectrometer conditions 3200 QTRAP equipment (quadrupole—linear ion trap), AB SCIEX, and electrospray ionization (Turbo Ion Spray) in negative mode. Appendix A compares the spectra. Significant changes in the identified ions in the specific mass-to-charge ratio (*m*/*z*) and the length of the bar, in both the control solution and the exposed solution, are evident.

Therefore, an improvement in the synthesis methodology was ensured, generating a material with better characteristics and performance than that obtained in the previous study [16,19]. Figure 2 represents the polymer structure and the mechanism of Pb^2+^ detection.

The obtained polymeric material IIP was evaluated using some characterization tests to verify and explain its properties. Figure 3 illustrates the FTIR spectra, where it is possible to identify the vibrational stretching signals of the C=O group of the cross–linking agent MBA at 1642 cm^−1^ [36], as well as the broad and intense band at an interval of 3100–3600 cm^−1^ in all the samples, which could be attributed to the superimposed absorption bands of the N–H and O–H functional groups of the secondary amide of the crosslinker MBA [28,36,37]. Figure 3 also shows a comparison of the IR spectra of the synthesized IIP materials IIP–AMPS–Pb (imprinted polymer with template), IIP–AMPS (imprinted polymer without template), and NIP–AMPS. In all materials, the polymerization occurred completely, since the vibration of the vinyl group C=C at 1620 cm^−1^ of the MBA disappeared in the polymer spectra [18,19]. The vibration at 1445 cm^−1^ in the group C–C and the band at 2932 cm^−1^ assigned to the group C–H with sp^3^ hybridization confirmed that the polymerization process was adequately completed [19,23].

SEM images of polymeric materials (Figure 4) showed heterogeneous particles with no defined shapes and with rough surfaces. Figure 4a,b illustrate the change in the texture of the surface of the material after the template leaching, showing a less compact and more porous texture (IIP–AMPS) compared with the non–leaching polymer (IIP–AMPS–Pb). The specific surface areas and porosities of the polymers demonstrate the mesoporosity of the IIPs as well as the large surface areas (see Appendix A).

The thermogravimetric analysis is shown in Figure 5. The figure illustrates that the TGA curves of all polymeric materials are composed of two stages of mass changes from the initial temperature to 800 °C. The first stage is from 37 °C to approximately 300 °C, with a weight decrease of approximately 20% assuming moisture loss. Furthermore, the second stage indicates the loss of total mass for IIP–AMPS. However, the IIP–AMPS–Pb polymer showed a remaining mass after 800 °C (approximately 10%), probably due to the greater strength of the chemical bonds in the polymer structure, which provided thermal resistance (See Appendix A). Similar behavior was observed in NIP–AMPS (Figure 5).

### 3.2. Adsorption Study

The adsorption capacity of polymeric material was explored, and the optimal conditions of pH and mass were determined. These were pH 6 and 10 mg, respectively, in agreement with preliminary studies [16,19]. The adsorption in IIP–AMPS reached a maximum at a pH = 6.0 ± 0.1, and this would be associated with the PZC of the polymeric material (close to 4, calculated experimentally) and the Pb^2+^ speciation (predominant ionic species at pH values < 7). At pH 6, the material surface would be negatively charged, facilitating an electrostatic attraction with Pb^2+^, and difficulties with the speciation of Pb^2+^ would be avoided. Additionally, it was determined that 10 mg of the polymeric material was the optimal mass necessary to achieve the highest percentage of adsorption at pH 6 in a 10 mL of 10 mg L^−1^ Pb^2+^ solution (Appendix A).

The adsorption study of Pb^2+^ was performed at predetermined optimal conditions: pH = 6, 10 mg of the polymer, and 10 mL of the adsorbate at an initial concentration of 10 mg L^−1^. The adsorption kinetics curves of IIP–AMPS and NIP–AMPS (Figure 6) show that imprinted materials achieved considerably higher adsorption than that of their counterpart nonimprinted materials (NIPs) in the first minutes of contact, reaching equilibrium conditions within a few minutes of contact (10 min). Therefore, 10 min was considered to be the equilibrium time for further experiments.

Kinetics models were applied to compare the experimental results obtained in the adsorption process with the pseudo-first-order model and pseudo–second–order model, in order to study the adsorption kinetics of Pb^2+^ ions on IIP–AMPS and NIP–AMPS materials. Appendix A show the kinetic parameters determined with the models for the polymeric materials. According to the values of the linear regression correlation coefficient (R^2^) and the error percentage of the theoretical q_e_, the best–fit model was a pseudo–second–order model (R^2^ = 1, error less than 0.5%), assuming an adsorption rate by chemical reaction mechanisms through the sharing or exchange of electrons between the adsorbate and adsorbent [38,39]. The fitting results are presented in Appendix A (nonlinear modeling) and Figure 6b (pseudo-second-order model linear modeling). The data in Table 1 show that the theoretical q_e_ (cal) values were closer to the experimental q_e_ (exp), with a 0.14 error for IIP–AMPS and 0.01% for NIP–AMPS. That indicates that the pseudo–second–order model predicted the experimental kinetics adsorption values of Pb^2+^ with good accuracy, suggesting that the adsorption rate of Pb^2+^ depends on the availability of adsorption sites rather than the concentration of the solution [16,38,40]. Additionally, the fast binding of the template ion achieved by the IIP–AMPS materials (10 min of contact) compared with its non-imprinted material, NIP-AMPS, could be attributed to the high-affinity recognition sites because the k_2_ values of the IIP were significantly higher than its corresponding NIP.

Adsorption isotherms were obtained by considering the adsorption capacity (q_e_) and concentration of the adsorbate (Ce) at equilibrium. The optimal parameters (pH 6.0 ± 0.1, 10 mg of material, and equilibration time of 10 min) were used to evaluate the adsorption capacity of the polymeric material in Pb^2+^ solutions with initial concentrations in the range of 0.5 to 100 mg L^−1^. Figure 7a presents the adsorption isotherms of IIP–AMPS and NIP–AMPS, where the curves show a slightly concave ascending shape. According to the L-type curve according to the Giles classification [41], these curves would indicate a progressive saturation of the active sites of the material without clearly presenting a limit of the adsorption capacity at high concentrations [39,40,41].

The adsorption capacity of IIP–AMPS demonstrated superiority over its non-imprinted material NIP–AMPS, reaching a maximum adsorption capacity of 54.11 mg g^−1^, in contrast to 29.26 mg g^−1^ of NIP–AMPS. In addition, isotherm models were applied to explain the adsorption process mechanism. Appendix A provides the correlation of experimental values with the isotherm models of Freundlich, Langmuir, Elovich, Temkin, Dubinin–Radushkevich and Redlich–Peterson (Appendix A). According to the statistical analysis (correlation coefficient and the Chi-square value (X^2^)), the Langmuir model fit the experimental data best (R^2^ > 0.99). The linear adjustment with the Langmuir model is depicted in Figure 7b; this would indicate that the adsorption process occurred due to the existence of a monolayer without competition between the molecules adsorbed at adjacent sites [39,40]. Linear adjustment with the Temkin model was performed; Appendix A reveals that this model did not fit linearly to the experimental values with a significant deviation of R^2^ = 0.931, which is why only the Langmuir model was chosen.

Furthermore, the applicability of the IIP–AMPS polymeric material was studied, and the selectivity against interferences Ni^2+^, Cu^2+^, Cd^2+^, Zn^2+^, Ca^2+^, Fe^2+^, and Hg^2+^ was evaluated. Appendix A summarizes the distribution coefficient values (K_d_, L g^−1^), selectivity coefficient k, and relative selectivity coefficient (K’). IIP–AMPSA has a high selectivity coefficient for Pb^2+^ concerning interfering ions, which indicates a great specificity in the adsorption process. Additionally, the relative selectivity coefficient (K’) of IIP–AMPS in general was higher than 8, which is higher than the value indicated in the literature (K > 1), suggesting a high selectivity towards the analyte [11,16]. The adsorption capacities of the materials for each ion are shown in Figure 7c for a better illustration.

The reuse of adsorbent IIP–AMPSA was confirmed. Figure 7d shows four adsorption/desorption cycles without a significant decrease in the adsorption capacity (less than 2% after four recycling tests); thus, IIP–AMPSA exhibited an excellent regeneration adsorption efficacy.

From the study of adsorption isotherms, the range of quantification and application of the colorimetric sensor was obtained, depending on the removal percentage. The range of concentrations with a removal percentage greater than 98% was considered to ensure the applicability and reliability of the sensor readings [20,22]. Therefore, the range between 0.5 to 10 mg L^−1^ was selected, and the color change in the solid was evident, as shown in Figure 8.

### 3.3. Colorimetric Sensor Implementation and SMARTPHONE Processing

The implementation of the colorimetric sensor consisted of the use of the previously synthesized polymer, which was exposed to solutions contaminated with the metal lead. Through an adsorption process under optimized conditions, the polymeric material removed the metal from the problem matrix, generating a color change in the solid (Figure 8). This characteristic was analyzed through a colorimetric study. For this, an image acquisition system coupled to a smartphone was implemented to obtain a photograph of the series of solids exposed to solutions of different and increasing concentrations. The obtained images were processed and analyzed using Image J 1.54g software and Trigyt (https://trigit.com.au/), a free web application [42], to obtain the RGB, YMK, and HSV color channels of each one. Finally, through a multivariate PLS analysis (using Unscrambler X 10.4 software), a linear correlation was obtained between the concentrations and the predicted values of the color channel system. From this, a calibration curve was generated to quantify real samples according to the change in the color of the polymer (exposed to solutions with the analyte).

The evaluation was carried out using the remaining solids of the Pb adsorption isotherm process in a concentration range of 0.5 to 10 mg L^−1^. They were worked with after a drying process in an oven at 60 °C, grinding, and weighing (approximately 5 mg of polymer powder). The collected solids were placed in the sample holder of the analysis system seen in Figure 1. The tests were carried out using the smartphone equipment. For this, the samples from the adsorption isotherm analysis were used. The radesphone accessory (under optimized lighting conditions) was used (Figure 1a) to photograph the solids; a noticeable color change from whitish to caramel was evident in the polymeric material. This coloring intensified as the Pb^2+^ concentration of the test solution to which the solids were exposed increased. Figure 9 illustrates a sequence of images of the solids in ascending order of concentration, noting an increase in the hue of each sample compared to the solid that was not in contact with a problem solution, and was defined as white (bk).

The images were processed in the ImageJ and Trigit programs, which evaluated the RGB, YMK, and HSV colorimetric channels, and numerical values were obtained by selecting a certain area to extract the average value of the recorded color intensities. Table 2 shows the extracted values.

#### Multivariate Analysis by PLS Regression

The multivariate analysis was developed with triplicate results of the standard solids with a concentration of 0.5 to 10 mgL^−1^ of Pb^2+^ (Table 2), employing Unscranbler X software. PLS, as a multivariate calibration method, reduces the dimensionality of the original variables in principal components to develop a regression model that provides the lowest prediction error [35]. Additionally, cross-validation is performed to evaluate the covariance between a set of predictor variables, X (color channel values), and response variables, Y (concentration). The model predicts the concentration of Pb^2+^ in different samples.

It was possible to obtain several graphs depending on the number of principal components, which provide various predictions with different values of the coefficient of determination R^2^ between the measured value and the predicted value. The higher value of R^2^ allowed for choosing the number of principal components, Figure 10 demonstrates a better prediction working with seven principal components in both calibration and cross-validation, indicating that seven components provide the best regression model (R^2^ = 0.998). Figure 11 shows the performance of the PLS regression model using seven principal components. The model fit the real and predicted values well, so it will be used for future sample predictions.

The repeatability of the proposed colorimetric sensor was evaluated using three solution samples of Pb^2+^ at different concentrations. The results using the prediction of the PLS regression with a coefficient of determination R^2^ equal to 0.998 are shown in Table 3. In addition, the limit of detection (LOD) was calculated based on the standard deviation of the regression of the PLS model between the measured and predicted values [43] according to Equation (2). The results presented in Table 3 show that, for the three samples analyzed, the relative standard deviation (RSD) was less than 5%; this points to a high degree of agreement between the measurements, indicating adequate repeatability. Moreover, the reusability study confirmed the reproducibility of the colorimetric sensor.

The results presented in Table 3 show that, for three samples, the relative standard deviation (RSD) was less than 5%; this points to a high degree of agreement between the measurements of the three fibers. The analytical curves related to the average reflectance of the fibers as a function of the concentration corresponding to these measurements were constructed. The LOD obtained was comparable with the LOD of the flame atomic absorption spectrophotometry (FAAS) analytical method. The sensitivity obtained with the colorimetric sensor indicated a lower detection sensitivity compared to other studies [29,30,31]; however, this option has the advantage of being a fast, reliable, convenient, non-destructive, and portable system, which makes it easy to handle when monitoring various matrices of interest and offers detection limits comparable with the standardized analysis method (FAAS).

### 3.4. Applicability

To evaluate the practical applications of the proposed system, real samples—including tap water; mineral water; and Paranapura, Huallaga River water—were quantified using inductively coupled plasma mass spectrometry (ICP–MS) Thermo Scientific iCAP RQ and the smartphone sensor with colorimetric detection. For a typical test, the real samples were the pretreatment. The suspended solids in the samples were removed firstly by being filtered through a 0.45 μm membrane, and then all the samples were subjected to an acid digestion process with microwave assistance in MARS6 digestion equipment (One Touch Technology) and acidified with nitric acid ultra-trace metal (>99.999%). All real samples were quantified using ICP-MS. Through multielemental analysis, some metal Cd^2+^, Cu^2+^, Hg^2+^, Zn^2+^, and Ca^2+^ (<50 ppb) and Pb^2+^ (<0.3 ppm) could be detected at trace levels. This result suggests that the concentrations of targeted metal ions in those samples were below the detection limit of the FAAS method, or there were no targeted metal ions in those samples at all. Therefore, some river samples were fortified with a patron solution of Pb^2+^ and then were subjected to smartphone sensor analysis and ICP-MS for quantitation. Table 4 lists the quantitation results and reveals that the recoveries determined by the proposed method (colorimetric sensor) were in the range of 85% to 102%. In contrast, ICP–MS measurements obtained a minimum of 98% of recoveries, with RSDs in the range of 1.12% to 8.20% for the same batch of samples. These results confirmed that the quantitation of smartphone-based colorimetric quantification and ICP–MS and FAAS showed no systematic difference, validating its applicability and reliability for on-site metal ion monitoring.

The results obtained in Table 4 show %RSD values less than 10% when comparing the concentration values determined by ICP–MS and by the proposed colorimetric sensor. The recovery percentages using the smart device indicate adsorptions greater than 90%, suggesting a minimal influence of the sample matrix. The difference between the proposed analysis method and the contrast analysis method (ICP–MS) was possibly due to the interferences present in the color analysis and the greater selectivity of the spectroscopic method. However, the intelligent device and the color analysis methodology with multivariate treatment offer the option of further use of polymeric materials in different systems and analyses, adding the characteristics of these materials to the color treatment technology through intelligent devices.

## 4. Conclusions

The present work reports the development of a colorimetric sensor based on intelligent platforms, polymeric material, and channel color value analysis for the quantitative detection of Pb^2+^ in water samples. The polymeric material was synthesized using the precipitation method; some parameters were optimized to achieve adsorption capacities higher than those obtained in a previous study. For this, an adsorption study of Pb^2+^ was carried out which revealed that adsorption mechanisms are associated with pseudo-second-order models and L-type Langmuir isotherms. It can be inferred that the adsorption process occurred in the active centers with a heterogeneous surface, resulting in adsorption without saturation at high concentrations due to chemical processes between the adsorbate and the adsorbent. Considering the favorable properties of the polymeric materials, the optimal parameters obtained in the adsorption study (pH 6, 10 mg IIP–AMPS, and 10 min of contact) were used in the implementation of the colorimetric sensor for the quantitative detection of Pb^2+^. The quantification method included the use of a smartphone camera to capture images of the reference samples (solids recovered from the adsorption isotherm study) using the device designed for this purpose. Subsequently, the images were digitized using Image J and Trigit software in different color spaces (RGB, I, HSV, w, and YMK). PLS multivariate regression determined the optimal working range (0–10 mg L^−1^), finding optimal values of the correlation coefficient R^2^ = 0.998 with seven principal components and a limit of detection (LOD) of 0.215 mg L^−1^. Finally, the applicability of the colorimetric sensor in real samples (river samples), based on multivariate regression, demonstrated a coefficient of variation of less than 6% in the determination of Pb^2+^ concentrations. Validation was performed using ICP-MS as a reference method. These results confirm that the proposed smartphone-based colorimetric sensor is a suitable analytical tool, providing a colorimetric quantification system with adequate selectivity that is non-destructive, portable, and easy to use for the reliable monitoring of Pb^2+^. Considering the results, this study would represent a pioneer in the application of ion–imprinted polymers incorporated into smartphone devices for the quantitative detection of heavy metals.

## Figures and Tables

**Figure 1 polymers-16-01523-f001:**
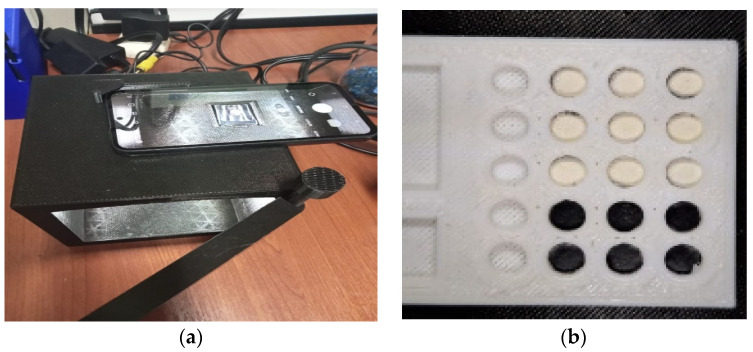
Image acquisition system (**a**) image capture accessory attached to the smartphone; (**b**) sample holder containing polymeric material.

**Figure 2 polymers-16-01523-f002:**
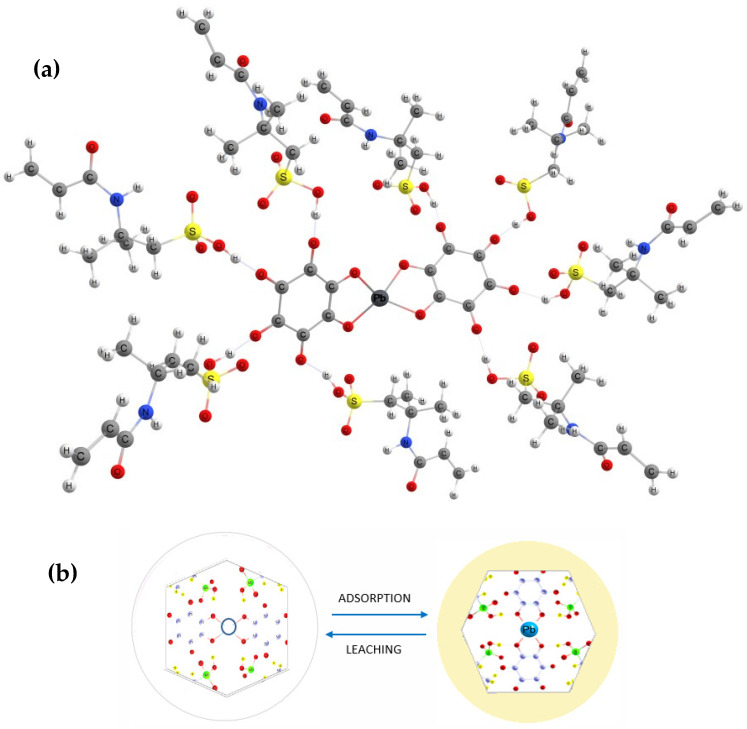
Representation of the polymer. (**a**) Basic structure of the polymer, including the interaction between Pb ion, ligand, and functional monomer; (**b**) colorimetric detection mechanism and the color change in the solid in the absence and presence of the Pb^2+^ ion.

**Figure 3 polymers-16-01523-f003:**
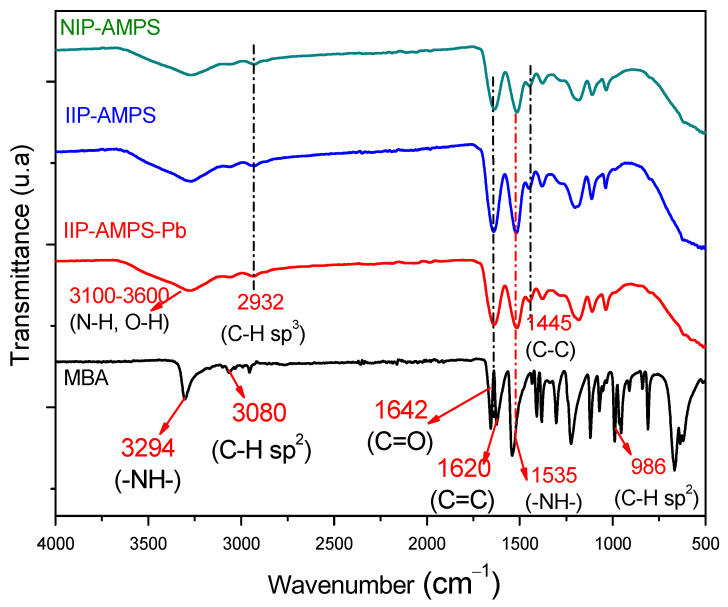
FTIR spectra of IIP–AMPS–Pb, IIP–AMPS, NIP–AMPS, and the crosslinker MBA.

**Figure 4 polymers-16-01523-f004:**
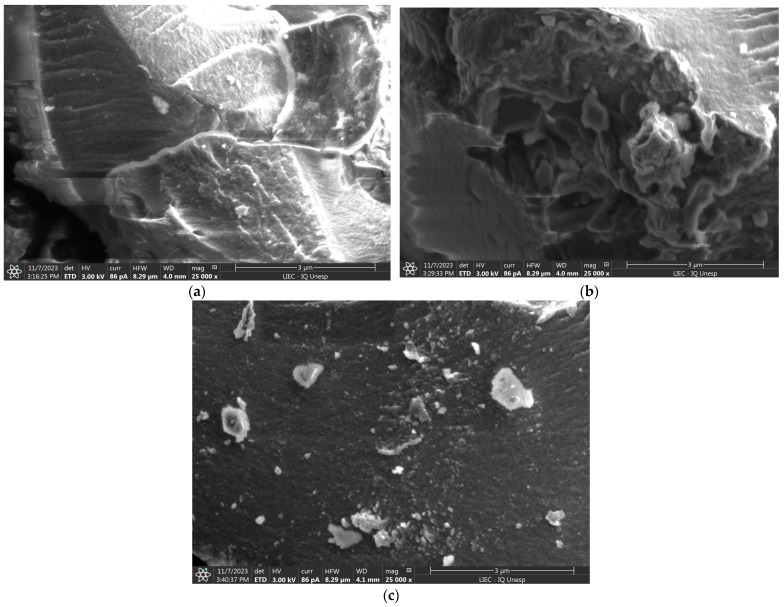
Characterization test. (**a**) SEM image of IIP–AMPS–Pb, (**b**) SEM image of IIP–AMPS, (**c**) SEM image of NIP–AMPS.

**Figure 5 polymers-16-01523-f005:**
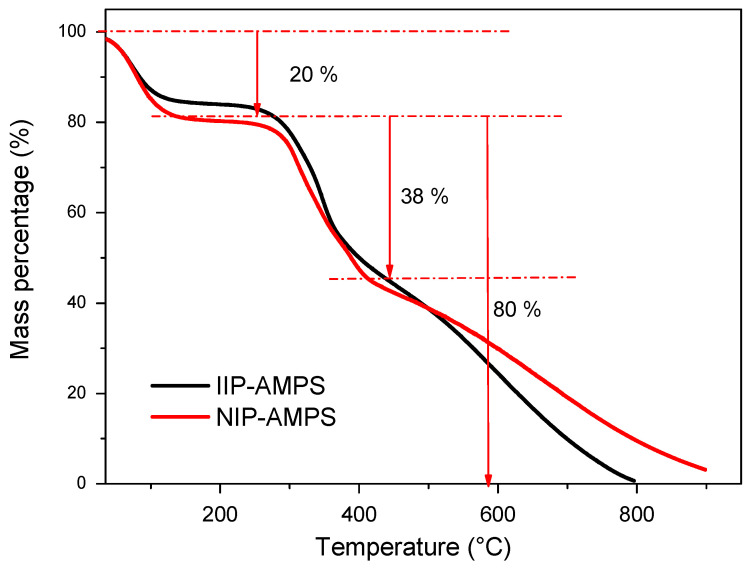
TGA graphs of IIP–AMPS and NIP–AMPS.

**Figure 6 polymers-16-01523-f006:**
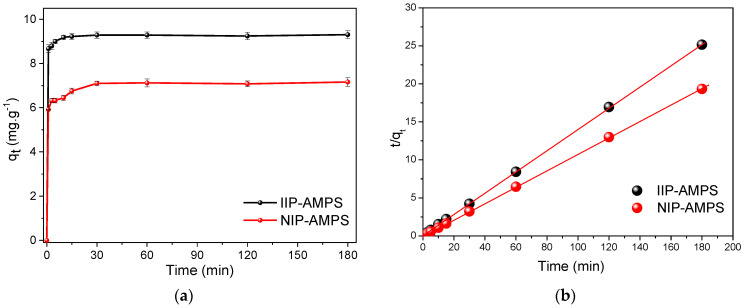
Kinetic study results. (**a**) Adsorption kinetics of Pb^2+^ using IIP–AMPS and NIP–AMPS, (**b**) linear adjustment with a pseudosecond–order model. Experimental conditions: pH = 6.0 ± 0.1, adsorbent mass = 10 mg, initial concentration = 10 ppm.

**Figure 7 polymers-16-01523-f007:**
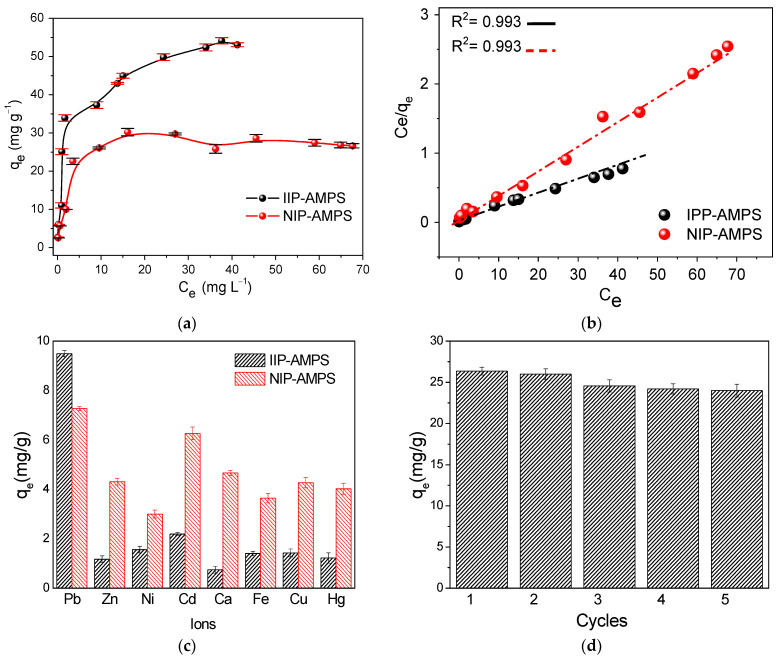
Adsorption isotherm study. *(***a**) Comparison of adsorption isotherms of IIP–AMPS and NIP–AMPS for Pb2+. (**b**) Linear adjustment with Langmuir model of the adsorption isotherm of IIP–AMPS and NIP–AMPS. (**c**) Selective adsorption of Pb^2+^ on IIP–AMPS. (**d**) Reusability of IIP–AMPS.

**Figure 8 polymers-16-01523-f008:**
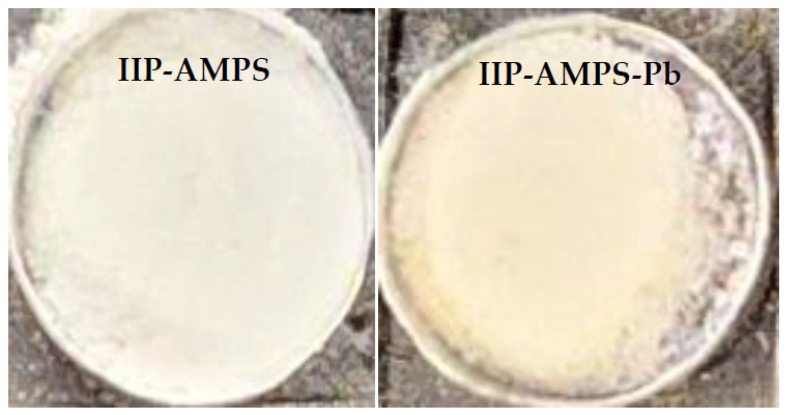
Color change of the polymer at the maximum adsorption concentration with more than 98% Pb^2+^ removal.

**Figure 9 polymers-16-01523-f009:**
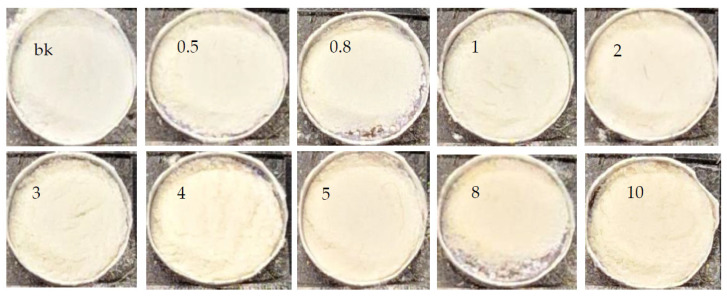
IIP photographs obtained using the smartphone, with a concentration variation of 0.5–10 mg L^−1^.

**Figure 10 polymers-16-01523-f010:**
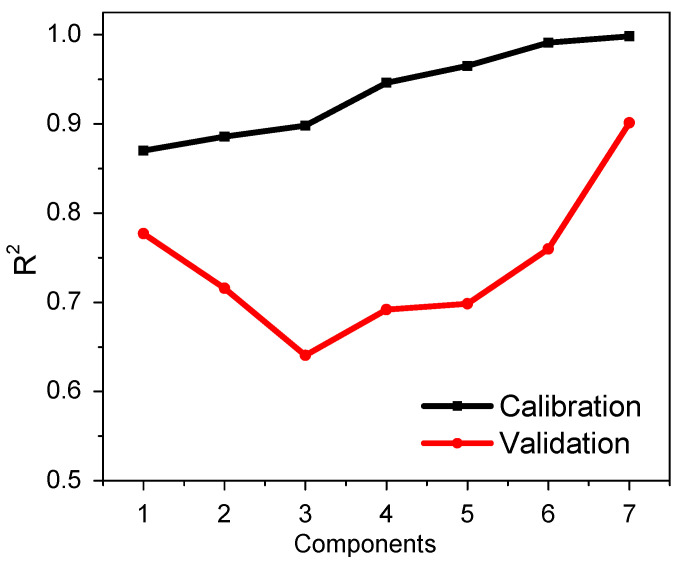
Correlation between the number of principal components with the R^2^ calibration and validation (n = 3).

**Figure 11 polymers-16-01523-f011:**
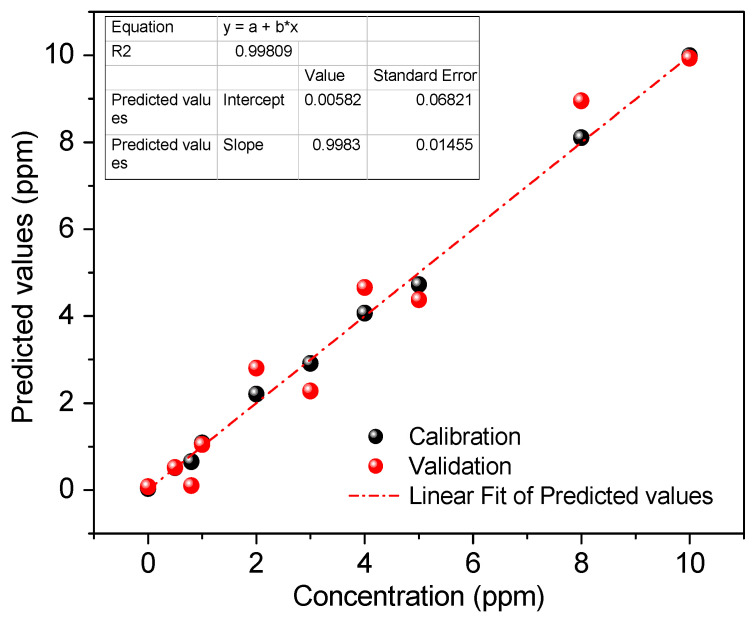
PLS regression model of real and predicted curve using seven principal components (n = 3).

**Table 1 polymers-16-01523-t001:** Correlation parameters for the pseudo–second–order kinetic models applied in the adsorption of Pb^2+^.

		Pseudo–Second–Order			
Sorbents	^1^ q_e (exp)_ (mg g^−1^)	^2^ q_e (cal)_ (mg g^−1^)	k_2_ × 10^−3^ (g mg^−1^ min^−1^)	R^2^	Error %
IIP–AMPS	9.31 ± 0.12	9.30	8.84	1.00	0.14
NIP–AMPS	7.16 ± 0.20	7.16	2.68	1.00	0.01

^1^ Adsorption capacity, experimentally determined; ^2^ adsorption capacity determined by modeling.

**Table 2 polymers-16-01523-t002:** RGB, YMK, HSV, and intensity values were obtained for the images of the polymeric solids exposed to standard solutions (0.5–10 mg L^−1^).

Standard (mg L^−1^)	R	G	B	M	Y	K	H	S	V	I
0.5	242.9 ± 0.8	233.1 ± 0.7	207.9 ± 0.7	4.0 ± 0.1	14.0 ± 0.9	5.0 ± 0.2	43.1 ± 0.8	14.5 ± 0.9	95.3 ± 0.6	228.0 ± 0.2
0.8	243.3 ± 1.2	232.7 ± 1.4	208.1 ± 0.6	4.0 ± 0.5	14.0 ± 1.0	5.0 ± 0.3	42.0 ± 1.2	14.5 ± 0.8	95.4 ± 1.5	228.0 ± 0.0
1	239.2 ± 1.0	228.3 ± 1.1	201.2 ± 1.3	5.0 ± 0.8	16.0 ± 0.5	6.0 ± 0.7	42.8 ± 0.9	15.9 ± 1.0	93.8 ± 1.3	222.9 ± 0.3
2	245.3 ± 1.1	230.9 ± 1.3	204.7 ± 1.1	6.0 ± 0.3	17.0 ± 0.6	4.0 ± 0.5	38.8 ± 1.1	16.6 ± 0.8	96.2 ± 1.2	227.0 ± 1.4
3	242.6 ± 1.5	231.3 ± 1.3	203.1 ± 1.0	5.0 ± 0.4	16.0 ± 0.7	5.0 ± 0.2	42.8 ± 1.2	16.3 ± 1.1	95.2 ± 1.3	225.7 ± 1.2
4	252.1 ± 1.6	238.6 ± 1.2	204.8 ± 0.8	5.0 ± 0.2	19.0 ± 0.4	1.0 ± 0.2	42.8 ± 0.9	18.8 ± 1.3	98.9 ± 0.4	231.8 ± 0.6
5	245.3 ± 1.2	228.1 ± 0.9	200.4 ± 0.8	7.0 ± 0.3	18.0 ± 0.5	4.0 ± 0.1	36.9 ± 1.3	18.3 ± 0.7	96.2 ± 0.9	224.6 ± 0.7
8	245.0 ± 1.2	224.5 ± 1.1	193.0 ± 1.1	8.0 ± 0.4	21.0 ± 0.5	4.0 ± 0.0	36.3 ± 0.9	21.2 ± 0.4	96.1 ± 0.9	220.9 ± 0.9
10	246.1 ± 0.9	227.7 ± 1.5	195.1 ± 0.9	8.0 ± 0.6	21.0 ± 0.1	3.0 ± 0.3	38.3 ± 1.0	20.7 ± 0.6	96.5 ± 1.0	223.0 ± 1.2

**Table 3 polymers-16-01523-t003:** Figures of merit for the PLS model developed for the colorimetric sensor for Pb^2+^.

Samples (mg L^−1^)	Repeatability (n = 7) Concentration Found (mg L^−1^)	RSD (%)	Limit of Detection (LOD) (mg L^−1^)
1	0.98	3.5	0.215
5	4.86	4.8	
8	8.01	4.6	

**Table 4 polymers-16-01523-t004:** Recovery percentages of Pb^2+^ obtained in real samples fortified with 1 and 5 mg L^−1^ using the smartphone-sensor and ICP–MS (experimental conditions: interaction time of 10 min; pH 6, n = 3).

Samples	Concentration Added (mg L^−1^)	Smartphone Sensor		ICP–MS		RSD (%)
		Concentration Found (mg L^−1^)	Recovery (%)	Concentration Found (mg L^−1^)	Recovery (%)	
Paranapura River	-	0.21 ± 0.02	96	0.23 ± 0.01	98	1.12
	1	1.05 ± 0.02	85	1.21 ± 0.02	98	7.08
	5	5.08 ± 0.03	97	5.22 ± 0.01	100	1.36
Huallaga River	-	0.23 ± 0.02	96	0.25 ± 0.03	104	4.17
	1	1.18 ± 0.01	95	1.22 ± 0.03	98	1.67
	5	5.01 ± 0.02	96	5.41 ± 0.02	103	3.84
Tap water	-	0.22 ± 0.03	96	0.228 ± 0.01	99	1.79
	1	1.16 ± 0.02	94	1.29 ± 0.02	105	5.31
	5	5.11 ± 0.03	98	5.30 ± 0.03	101	1.83
Well water	-	0.28 ± 0.01	85	0.33 ± 0.02	100	8.20
	1	1.36 ± 0.04	102	1.52 ± 0.01	99	5.56
	5	5.09 ± 0.02	95	5.12 ± 0.02	98	1.45

## Data Availability

The raw data supporting the conclusions of this article will be made available by the authors on request.

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
