# Peer review of "Smartphone-Based Rapid Quantitative Detection Platform with Imprinted Polymer for Pb (II) Detection in Real Samples"

_polymers, 2024, doi:10.3390/polym16111523_

Round 1

Reviewer 1 Report

Comments and Suggestions for Authors

The work by Rosario López and co-authors is aimed to the development and application of an efficient method for quantifying Pb2+ in aqueous samples using a smartphone-based colorimetric device with imprinted polymer (IIP). This report is well written and it contains the main scientific aspects including fabrication of sensor materials, studying its absorption kinetic, and finally the determination of lead ions in water probe using a self-made smartphone-based colorimetric device. This work contains elements of novelty (development of sensor material and studied it adsorption kinetic) and practical significance (the determination of leads ions in the real water probes using the developed sensor). The verification of obtained results of lead ions determination with the use of alternative method is emphasized the reliability of the proposed sensor. I believe it can be accepted with minor changes.

1.       Along with the most common water pollutants (Pb, Cd ions) the Hg is often existed in rivers. Why did authors exclude it at selectivity tests?

2.       The ability of sensor to determine lead concentration in the water in wide pH range (5-7) is important for its application. Although the optimal parameters of lead ions detection using developed sensor have been found, I wonder would this sensor determine the content of lead ions correctly, under others conditions, for example, at pH=5.

3.       The adsorption isotherm of IIP-AMPS has a two-stage type. Did the monolayer adsorption models are applicable for interpretation of adsorption isotherm of IIP-AMPS? Why did the BET model excluded?

4.       Authors reported that sensor can be reused.  How has the reversible response of sensor been achieved? Please clarify.

Reviewer 2 Report

Comments and Suggestions for Authors

Excellent work

Author Response

We are grateful for your insightful comments and deeply appreciate the referee's dedication in reviewing our manuscript

Reviewer 3 Report

Comments and Suggestions for Authors

This article reports on the detection of lead ions using smartphone cameras. This paper is expected to contribute to the development of the field.

Generally, such detection methods are carried out with the naked eye, but why use a smartphone camera? Isn't the naked eye a better system than a smartphone camera?

The Samsung smartphones used in this study are thought to be new models, but what level of camera performance is required to use them for this detection method?

Is a darkroom necessary for this assay? Wouldn't it be better to be able to detect the metal ions of interest in real-time outdoors and in sunlight?

Please provide the structural formula of the polymer used in this study. Also illustrate by what method the target metal is detected.

It is well written and worthy of acceptance if the sections I commented on to the author can be corrected.
